# "It let me merge my love of teaching with research": A qualitative investigation of the career pathways of biology education researchers

Emily P. Driessen[1,2]*, Ariel L. Steele[1,2], Robin A. Costello[1], Peyton Brewer[1], Cissy J. Ballen[1]

1 Department of Biological Sciences, Auburn University, Auburn, Alabama, United States of America,
2 Department of Biology Teaching and Learning, University of Minnesota, Minneapolis, Minnesota, United States of America

* Dries046@umn.edu

**Data Availability Statement:** The data analyzed in this study describe the career paths of a sample of

## Abstract

Discipline-based education research—a field of research that investigates teaching and learning within STEM disciplines—has emerged over the last few decades to improve the quality of STEM education worldwide. Simple qualitative questions concerning the career backgrounds and motivations of the individuals who conduct this research have yet to be explored. Here, we surveyed and interviewed discipline-based education researchers about their career trajectories and motivations to pursue this field of research. We focused specifically on recruiting biology education research faculty members at colleges and universities. We used the Social Influence Model and Social Cognitive Career Theory to develop and analyze survey and semi-structured interview questions. Findings revealed participant career paths all began with disciplinary undergraduate and graduate-level biology education. We noticed participants began conducting biology education research due to their *values* and *personal interests*, while additionally being swayed by *contextual factors*. Specifically, participants *valued* biology education research because it allowed them to make a difference in the world and provided them with a community open to change and collaboration. Biology education research allowed them to explore their *interests* in teaching and evidence-based approaches to education. These *values* and *interests* were impacted by *contextual factors*, including discoveries of opportunities, positive (or negative) experiences with mentorship, exposure to evidence-based teaching literature, considerations of salary and job security, and experiences with gender-based discrimination. Our results underscore the importance of harnessing individual values and interests—especially those centered on evidence-based teaching practices and making a difference in the world—while fostering a positive and supportive academic environment. This research reveals pathways toward discipline-based education research careers. Additionally, this research can inform the development of graduate programs and funding opportunities.

biology education researchers. The data includes enough demographic information that re-identification of individuals may be possible. For these reasons, it is not possible to publicly release the data. Qualified researchers may request access to data through Auburn University's Senior Vice President for Research and Economic Development, John Eric Miller (email: jem0068@auburn.edu).

**Funding:** We acknowledge the Division of Undergraduate Education (DUE) at the National Science Foundation (NSF) for providing funding to CJB (NSF-DUE 2011995). The funders had no role in study design, data collection and analysis, decision to publish, or preparation of the manuscript.

**Competing interests:** The authors have no competing interests to disclose.

## Introduction

Frequent calls advocate for improved science, technology, engineering, and mathematics (STEM) education to meet the objectives of increasing national science literacy [1–5]. Specific recommendations promote "student-centered classrooms" [6] and the adoption of "evidence-based teaching practices" [3]. These recommendations are supported and commonly investigated by individuals formally described as science faculty with education specialties (SFES) [7].

SFES are broadly defined as "scientists who take on specialized science education roles within their discipline" [8] and include those trained primarily as disciplinary scientists with expertise in K–12 science education, undergraduate science education, and/or science education research [7, 9]. These education specialists "advance science education by affecting undergraduate science education, influencing K–12 science education, and/or conducting discipline-based research in science education" [10]. Research describing SFES has focused on these positions in the context of biology, chemistry, geoscience, and physics departments [7–15]. SFES occupy variable types of positions that differ in tenure-status, teaching load, research responsibilities, independence, and relation to other faculty [7, 16]. Here, we build on prior research concerning SFES broadly, focusing on a smaller but growing subset of SFES: the discipline-based education research community.

Discipline-based education research investigates teaching and learning within STEM disciplines from a perspective that reflects the discipline's priorities, worldview, knowledge, and practices [17]. The goals of discipline-based education research are to: (1) understand how people learn science and engineering concepts, practices, and ways of thinking; (2) understand the expertise of a discipline and how that expertise is developed; (3) identify and measure pertinent learning objectives and instructional approaches that lead students toward those objectives; (4) contribute to the knowledge base in a way that translates to classroom practice; and (5) make science and engineering education more broad and inclusive [17]. While there is much to learn and apply across the discipline-based education research fields, (e.g., biology, chemistry, physics), each subdiscipline has its own culture and is in a unique stage of development.

Biology education research is one subdiscipline of discipline-based education research that has flourished nationally with opportunities to occupy tenure-track faculty positions, pursue PhD programs, and attend national conferences [17]. While some elements of the field of biology education research have been explored (e.g., how biology education research has matured over time [18], how biology education research can become more inclusive and anti-racist [19], and how members of the biology education research community conceptualize active learning in biology [20]), many questions concerning the nascent field of biology education research remain. For example, why and how do individuals initially pursue biology education research? To our knowledge, this study is the first to explore *what factors (i.e., values, personal interests, and contextual) led participants to pursue biology education research*?

We conducted a qualitative research study to address this question. Specifically, we designed and distributed an open-ended survey to faculty members who conduct biology education research. The survey inquired about participants' career trajectories and goals, motivations to pursue biology education research, perceived differences between the fields of disciplinary biology and biology education research, and identities and personal/professional values (S1 Table). While we acknowledge that biology education research is also a discipline in science, we use the term 'disciplinary science' and 'disciplinary biology' to refer to traditional scientific and biological research (e.g., ecology, genetics, biophysics). Hereafter we use these terms, in contrast to biology education research, which is a relatively new sub-discipline with increasing presence across several science fields. Next, we selected a subset of survey

respondents to participate in individual, in-depth semi-structured interviews. We asked interview participants open-ended questions like those on the survey (S2 Table); however, this additional method allowed us to ask follow-up questions in the interviews to probe for further information. To answer our research question, we used a combination of two theoretical frameworks and first and second cycle coding to analyze the survey responses and interview transcripts [21].

## Theoretical frameworks

In this present work, we drew from two frameworks, the Social Influence Model [22] and Social Cognitive Career Theory [23], to understand the career decisions made by faculty members who conduct biology education research. Applying these frameworks, we highlighted the values, personal interests, and contextual factors that impacted faculty career decisions and trajectories. Here, we describe the Social Influence Model and the Social Cognitive Career Theory to illustrate the motivating variables we captured in this study.

The Social Influence Model consists of three social influence processes that impact career motivations: (1) rule-oriented (or compliance-based) self-efficacy, (2) role identification, and (3) values [22]. For our purposes, we focused only on *values* from the Social Influence Model because the faculty members in our study already identified as scientists and internalized scientific values, and self-efficacy has previously been shown to poorly predict intentions to pursue a scientific career when participants already identified as scientists and internalized scientific values [22]. Values refer to both what is important to the faculty member and the perceived values of the academy. The Social Influence Model posits that it is important to consider the extent to which scientists make decisions about their interest in research and academia based on their values and the extent to which their values align with those held by the academic culture.

Like the Social Influence Model, Social Cognitive Career Theory posits career motivations develop from personal decisions made within social and institutional contexts [23]. However, Social Cognitive Career Theory expects *personal interests* lead individuals to pursue a particular training path while also recognizing *contextual factors*, specifically in the form of perceived barriers to career development [23–26]. In addition to the perception of barriers, Social Cognitive Career Theory contends that individuals' self-efficacy for coping with barriers is important to explore, just as stated by the Social Influence Model. However, again, given that previous literature showed self-efficacy poorly predicted intentions to pursue a scientific career when participants already identified as scientists and internalized scientific values, we do not focus on self-efficacy [22]. Taken together, we draw upon *values* from the Social Influence Model and *personal interests* and *contextual factors* from the Social Cognitive Career Theory. We used these three constructs to inform our qualitative analysis, so we can understand how these constructs played out in the participants' decision making about their career paths.

## Methods

This qualitative exploratory study uses open-ended survey questions and follow-up interviews to investigate the lived experiences of a sample of biology education research faculty members. While validity, reliability, and trustworthiness are standards for research [27], validity and reliability are often terms used when discussing quantitative research [27, 28], whereas trustworthiness is often used when discussing qualitative research [27–31]. Since the present study is qualitative, we do not present measures of validity or reliability. Rather, our methods and analyses focus on establishing trustworthiness throughout. Trustworthiness concerns the degree to which the qualitative data, analysis, and interpretations are true and rigorous [27, 32, 33].

More specifically, trustworthiness in qualitative research refers to the credibility, dependability, confirmability, transferability, and authenticity of the research [27]. We used various methods to establish trustworthiness throughout this study, including closely aligning our specific research question with our framework and experimental design [34], iteratively revising our open-ended survey instrument based on expert feedback [35], conducting interviews with a subset of our survey participants to expand on their survey responses [36], and inviting interview participants to perform member checking of our interpretations of their interview data [37, 38]. Importantly, the aim of this study is not to find a representative case that generalizes findings to the experiences of other Biology Education Research Faculty or Science Faculty with Education Specialties. Rather, the aim of the present study is to develop an understanding of the career motivations and career timelines of the participants who agreed to be in this study [39].

## Positionality

Each author brings a variety of identities and experiences to the team, and these interact with all stages of the research process [40]. All the authors are white women who have backgrounds in disciplinary biology and conduct biology education research. Here we detail each of our paths to biology education research as they are relevant to the current study.

The first author (EPD) is a postdoctoral researcher in the department of Biology Teaching and Learning at the University of Minnesota, studying inclusive teaching strategies in undergraduate biology courses. At the time of this study, she was in her final year of earning a PhD in biology while researching the efficacy of active learning strategies in undergraduate biology courses. Prior to that, she earned a master's degree in science, technology, engineering, and mathematics (STEM) education and a bachelor's degree in microbiology.

The second author (AS) is a postdoctoral researcher in the department of Biology Teaching and Learning at the University of Minnesota. At the time of this study, she was in the final year of earning a PhD in higher education and researching women's experiences in graduate STEM education. Previously, she obtained a master's and bachelor's degree in biology.

The third author (RAC) is, and was at the time of this study, a postdoctoral researcher in the Department of Biological Sciences at Auburn University conducting research on the effect of counter-stereotypical scientists on student outcomes in biology. She received a PhD in evolutionary biology, studying social interactions in beetles, and a bachelor's degree in biology.

The fourth author (PB) is, and was at the time of this study, an undergraduate student in the Department of Biological Sciences at Auburn University, and she conducts biology education research with the other authors.

The senior author (CJB) was, at the time of this study, an assistant professor of biology education research in the Department of Biological Sciences at Auburn University. Previously, she conducted biology education research in two postdoctoral positions and as an undergraduate research assistant. She obtained a PhD in evolutionary biology, studying the evolution of color signaling in reptiles, and a bachelor's degree in wildlife and biology.

In all our career paths, we started in disciplinary biology and transitioned to biology education research. We acknowledge that our pathways to biology education research are unique and informed our decision to conduct this study. They also have the potential to influence how we interpret participant's responses, many of whom had similar paths to our own. In some ways this may strengthen our conclusions but could also lead us to inadvertently overlook other potential sources that drove people into biology education research.

## Participant recruitment

To recruit a national sample of teaching and research faculty who conduct biology education research, we used three methods between August 31st, 2021 and December 15th. All three recruitment methods involved using the same online Qualtrics survey (see Survey Development Section for survey specifics). First, we emailed the study information and Qualtrics survey link to the listserv for the Society for the Advancement of Biology Education Research (SABER), the world's largest scientific community of discipline-based education researchers and teaching practitioners who focus on improving postsecondary biology education through evidence and theory (https://saberbio.wildapricot.org). Second, we collected contact information for the personnel listed for all graduate programs in biology education research on SABER's website. Using this information, we emailed the study information and Qualtrics survey link to each person directly, attempting to recruit them for our study. Third, each of the authors posted the study recruitment information and Qualtrics survey link on our science Twitter profiles, encouraging our networks to both participate in the study, if eligible, as well as share the information with their networks. While there are certainly additional recruitment methods we could have used (e.g., emailed additional listservs), the intent of our study was not to generalize about every individual who conducts biology education research, but rather to begin to understand the motivations of a sample of individuals to pursue biology education research. Recruiting individuals from SABER's listserv and listed graduate programs was an appropriate starting point.

These recruitment methods allowed us to survey a sample of teaching and research faculty across the nation who conduct biology education research. Additionally, the Qualtrics survey included an option for participants to indicate their interest in a follow-up interview (i.e., "yes" or "no") (see Interviews section for interview specifics). This allowed us to contact participants who indicated their interest in participating in a follow up interview via email.

Due to the nature of the research (i.e., virtually recruiting participants), we were unable to obtain any "wet signatures." For this reason, potential participants were prompted with an information letter at the beginning of the Qualtrics survey which explained the scope of the research. At the end of the letter, participants had the option to click "I CONSENT," or "I DO NOT CONSENT" in the Qualtrics site. They also had the option to close the survey or not open it in the first place. The same process was repeated for participants who elected to participate in the Zoom recorded follow-up interview. We conducted all research in accordance with the Auburn University Institutional Review Board (Auburn IRB protocol no. 21–354 EX 2108).

Using these methods, we successfully recruited a sample of 26 participants for our survey, and a subset of six survey respondents consented to and participated in a follow-up interview. While we collected race and gender identity demographic information for each of the participants, using the online survey, we elected to omit this demographic information to decrease the chances of participant identification.

In recruiting participants with these three methods, we used both purposive and convenience sampling methods [41]. Purposive sampling involves recruiting participants with particular knowledge or experience (i.e., faculty members who conduct biology education research) [41]. Convenience sampling involves ease of recruitment, and our use of SABER's listserv allowed us to easily contact and recruit members of SABER, who were also likely to participate given both their familiarity with our research team members (i.e., all researchers aside from PB are members of SABER) and their dedication to improving biology education at all levels. Importantly, both purposive and convenience sampling methods involve setting a number of participants to recruit a priori [41]. However, we did not determine a number in

advance of the study. This is because (1) we were interested in myriad experiences of faculty members who conduct biology education research (i.e., as many as would participate) and (2) faculty members are often difficult to recruit due to their busy schedules. Instead, we decided on a length of time during which we were able to continue to recruit participants. We decided on a time span of four months due to researcher availability. Relatedly, we did not focus on data saturation (i.e., a practice that justifies stopping data collection once the same codes have come up repeatedly in a dataset) [42]. We made this decision because our research focus (i.e., the experiences and career paths of biology education research faculty) acknowledges the participants have different career paths and each person's exact experience is unique [42].

## Survey development

To investigate the factors (i.e., values, personal interests, and contextual) that led our participants to pursue biology education research, we designed a survey with open-ended response questions. Specifically, three of the researchers contributed to the development of survey items over the course of several meetings held over six months. Due to the nature of the survey, we checked it for "threatening questions," as social norms may influence how participants respond [43]. As recommended in prior literature, sensitive items were placed toward the end of the survey, and we administered the survey online and anonymously to increase both response rate and honesty of responses [43].

Following the survey's initial development, the survey went through multiple rounds of revisions. While it is standard practice to determine validity and reliability for survey questions that present a participant with a finite set of response options (e.g., Likert-scale survey questions) [44], these same standards do not apply to open-ended questions because they allow the participant an infinite and open-ended response. Instead, to establish trustworthiness, it is standard and appropriate to have several stages of review of the open-ended survey questions prior to data collection. First, the senior author, a biology education researcher and faculty member, took the pilot survey. Based on her responses and feedback, three of the other researchers refined the survey items. Then, they presented the items to a biology education research lab group, taking detailed notes and feedback and making suggested edits. This resulted in a final survey with 11 open-ended questions centered on career motivations and experiences in Biology Education Research (S1 Table), prompts to collect demographic information (i.e., gender identity and race/ethnicity), and a space for the participant's contact information for those interested in a follow-up interview.

After we solidified the open-ended survey questions, we created an online survey using Qualtrics survey software (https://www.qualtrics.com/). The survey (S1 Table) included the open-ended survey questions, as well as an information letter calling for participation, "in a research study to investigate your career motivations as a biology education research (BER) faculty member at a college or university." To ensure that all the participants who completed the survey were in fact conducting biology education research as faculty members, we additionally included the following survey question inquiring about the research conducted in their lab: "As a faculty member, broadly, what is the scope of the research conducted in your lab?" We checked participant answers to this prompt, and all the participant answers focused on biology education research as at least one type of research conducted in their lab. We distributed this survey to participants using a unique survey link.

## Interviews

Using email addresses obtained from the surveys, we contacted nine of eleven participants who indicated their interest to participate in a follow-up interview. We selected these nine

participants based on their varied and detailed responses to the survey prompts. Six of these participants scheduled, consented to participate in Zoom recorded interviews (see participant recruitment section for further information), and participated in interviews.

Four members of the research team conducted the interviews in pairs (i.e., EPD & PB; AS & RAC). Each of the pairs had an experienced interviewer (i.e., EPD, AS) and a researcher who has not previously conducted interviews (i.e., RAC, PB). Specifically, EPD and AS each had experience conducting more than 50 semi-structured interviews, and both were in their final year of their PhD programs. Due to this experience, EPD and AS each trained the other member of their pair on how to conduct interviews (e.g., follow the interview protocol, ask follow-up questions that are relevant to the research questions and research framework) and co-interviewed the participants. Each pair of interviewers conducted half of the interviews (i.e., 3 interviews for each pair). See positionality statement for further credentials at the time of the study.

We interviewed the participants using a semi-structured interview protocol designed to explore participants' career trajectories and goals, motivations to pursue biology education research, and their visions of how biology education research aligned with their identities and personal/professional values (S2 Table). The interview prompts and questions were similar to those used in the survey; however, the nature of the semi-structured interview protocol allowed us to inquire for more information, focusing on the Social Influence Model [22] and Social Cognitive Career Theory [23]. We conducted all interviews online using Zoom (zoom. us). Each interview took approximately one hour. We recorded the interview video and audio using Zoom (zoom.us). We additionally used Otter (Otter.ai) to both record a back-up copy of the audio and provide audio transcripts for each interview. After we finished recording, we checked each of the Otter transcripts and made corrections where there were discrepancies between the transcript and the audio recording. Of note, the pseudonyms and pronouns which we use throughout this study were selected by each of the interview participants.

## Data analysis

We describe the techniques we used to analyze the survey and interview data below, each in their own respective subsection. We used first and second cycle coding to analyze both the open-ended survey response text and the interview transcript text [21]. We describe this for each specific data type respectively.

**Survey analysis.** One month after recruiting participants to complete the surveys, we downloaded all survey responses. Three of the researchers read through answers to all eleven questions and focused on the questions and responses that were both the most detailed and most aligned with our frameworks (i.e., Social Cognitive Career Theory and the Social Influence Model). The three questions are highlighted in S1 Table. We assigned pairs of authors to review the responses and subsequently develop codes for one or two of these highlighted survey questions using first and second cycle coding [21].

For our first cycle of coding, we used in vivo coding, a process that uses the participants' own words (e.g., a single word or phrase) to generate codes names [21]. This coding process was additionally guided by the three constructs from our frameworks (i.e., interests, values, and contextual factors). Specifically, two of the researchers reviewed and coded responses for the first highlighted question: (1) Why did you change careers from disciplinary science to pursue biology education research? Separately, two of the other researchers reviewed and coded responses for the remaining highlighted questions: (2) How does discipline-based education research align with your values? and (3) What motivated you to pursue biology education research? After the two members of each team individually reviewed responses, developed codes, and coded the responses, they met with their partnering team member to discuss the

codes they created. After each pair created a complete coding rubric for each highlighted survey question, each member of the pair used their codebook(s) to individually code the question responses. Subsequently, they met again with their partnering team member and came to a consensus for each response [45]. Finally, the two sets of pairs met altogether to discuss the codes across the survey questions, looking for similar codes and ideas to streamline and lightly edit for consistency. This process led to the development of three final codebooks, one for each highlighted question.

For our second cycle of coding, we used focused coding to determine which codes were the most salient and frequent [21]. Specifically, each pair of researchers determined which codes they marked the most frequently and aligned these with the constructs (i.e., interests, values, and contextual factors) from the frameworks used in our study [21]. Notably, three codes appeared in all three codebooks (i.e., making a difference, for the love of teaching, evidence-based approaches to biology education) and two codes appeared in two of the three codebooks (i.e., burnout and best of both worlds). Additionally, there were 13 total unique codes generated by this process. To both avoid redundancy and reduce the number of codes, we focused our analysis only on the five most mentioned survey codes across all three codebooks. These included "making a difference," "for the love of teaching," "evidence-based approaches to education," "best of both worlds," and "discoveries of opportunities." For explanations of these codes, how each of these codes align with the constructs obtained from our two frameworks (i.e., Social Influence Model and Social Cognitive Career Theory), and a participant example for each code, see the light orange boxes in Fig 1.

**Interview analysis.** After we created the final codebook for our survey responses (Fig 1, light orange), we combed the interview transcripts for these same codes while additionally reading through each interview independently and noting new codes using in vivo coding for the first cycle of coding [21]. We also noted key events that contributed to participants' career paths toward a career in which they conduct biology education research. Then, all four coders (EPD, RAC, AS, & PB) met to discuss these codes and key events. Each coder then returned to the interview transcripts to use focused coding during the second cycle of coding to decide which codes were most salient to answer our research questions [21]. All coders met one final time to discuss the most salient codes, and coders discussed until reaching consensus. This resulted in the final codebook (Fig 1) with codes developed uniquely from the interview transcripts (Fig 1, dark orange) and codes developed from the survey analysis (Fig 1, light orange) that were then applied to the interview transcripts. The codes contributed from the interviews include: "change and collaboration," "mentorship," "the role of literature," "economic drivers," and "experienced sexism in disciplinary science." For explanations of these codes, how each of these codes align with the constructs obtained from our two frameworks (i.e., Social Influence Model and Social Cognitive Career Theory), and a participant example for each code, see Fig 1.

To illustrate the key professional stages in interviewed participants' education or professional careers, we created timelines for each participant using a diagram application called Lucidchart (https://lucid.app) (see Fig 2 for example; S1–S5 Figs for timelines of each interview participant) with salient excerpts from the interview transcripts attached to each stage. Finally, to focus solely on the type of training each participant engaged with at each education or professional career stage, we removed the excerpts from the figures and simply compiled all timelines into one graphic. To do this, we used BioRender, a tool used to create and share scientific figures (https://biorender.com). In these figures, we highlighted disciplinary biology research training in green and biology education research training in orange.

After we completed our analysis of the interviews, we created personalized manuscripts for each of the interviewed participants. These manuscripts highlighted the excerpts for each participant along with the claims we made using those excerpts. We then sent each participant

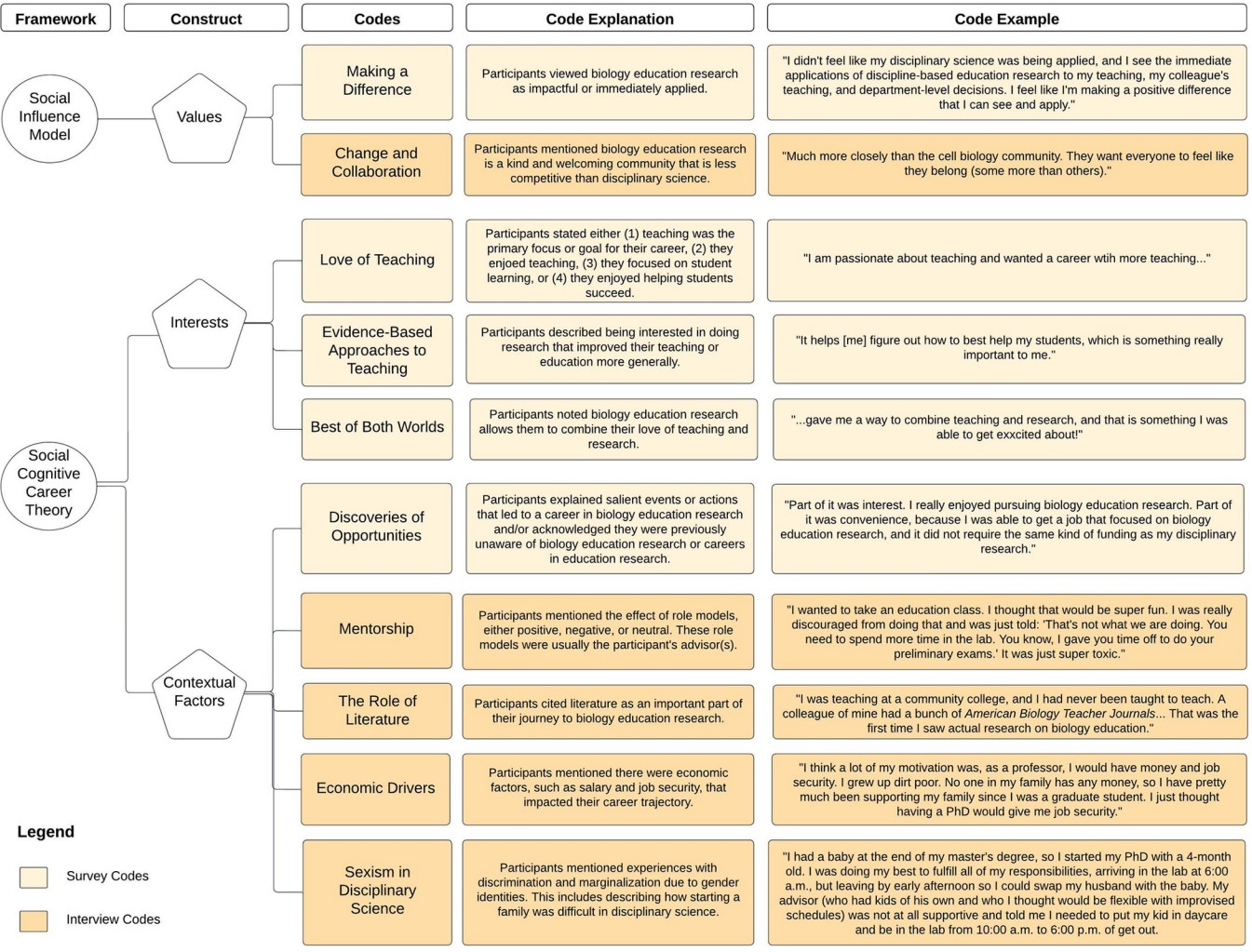

**Fig 1. An outline of the corresponding frameworks, constructs, and codes used to analyze both the surveys and interviews.** The code information includes explanations of each code as well as a participant example of each code. The codes in the light orange were created from the survey responses, and the codes in the darker orange were obtained from the interviews. Some of the survey codes, that were not among the top five most common, came up in the interviews and marked an important event for the participant. In this case, we attributed the code to being contributed by the interviews.

their personalized manuscript along with their career timeline, and we asked them to provide any alterations or edits to our interpretations of their experiences and stories. After members provided any alterations or feedback, we made the suggested changes to our analysis and interpretation. This is a practice called synthesized member checking, which provides an additional measure of trustworthiness [37].

## Results and discussion

From the data we obtained from the surveys and interviews, we noticed that participants had variable career paths and motivations that eventually led them to careers as biology education research faculty members at a college or university (Fig 3). In our sample, all participants came from science backgrounds. They all received graduate-level training in disciplinary science but began to conduct biology education research during or after graduate school. While participants held a variety of motivations for pursuing biology education research, they ultimately

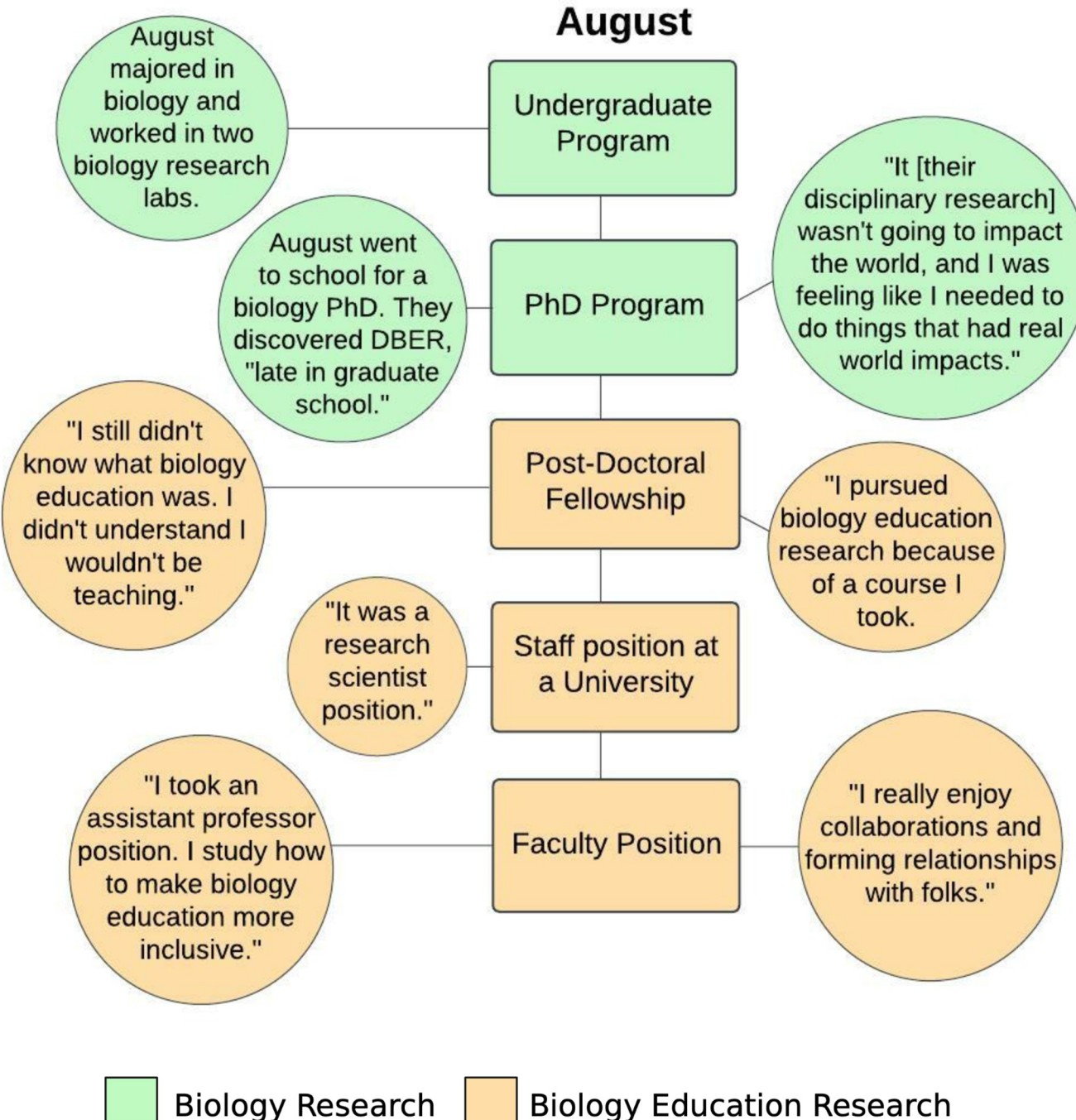

**Fig 2. An example timeline for the participant "August".** The timeline highlights key professional stages in the interviewed participant's education and professional careers. Green rectangles and circles represented experiences with disciplinary biology research. Orange rectangles and circles represent experiences with biology education research.

began conducting biology education research due to their *values* and *personal interests*. They were additionally swayed by *contextual factors*. We explain these factors and their corresponding codes, support them with survey and interview excerpts, and place them in the context of the literature in the following sections.

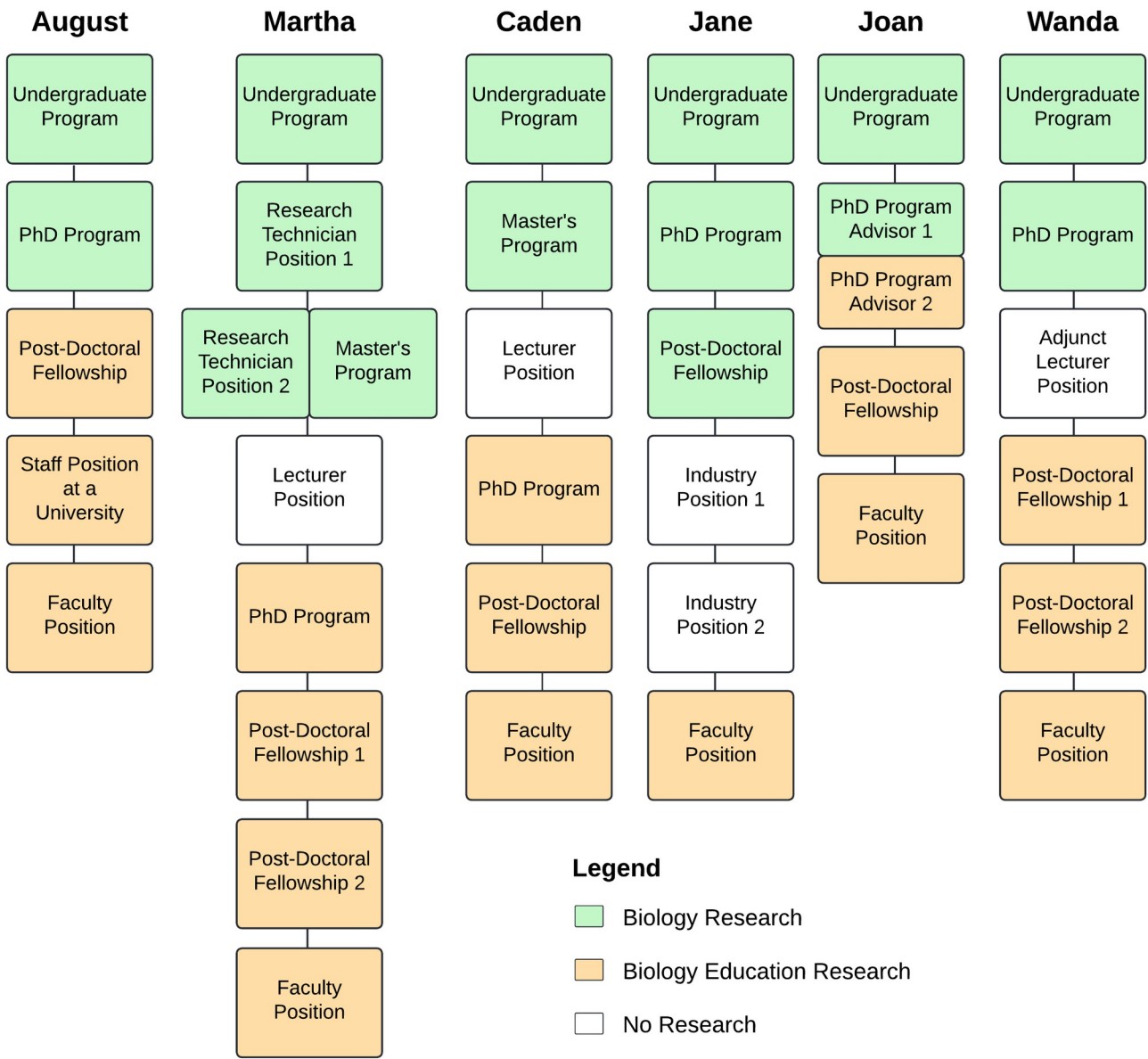

**Fig 3. The career pathway for each of the interview participants.** Green rectangles represent a professional experience in disciplinary biology, orange rectangles represent a professional experience in biology education research, and white rectangles represent professional experiences that were not focused on research.

## Values

In line with the Social Influence Model, participants held values that played a role in their eventual participation in biology education research [22]. These values included "making a difference" in the world and the desire to work in a community that is open to "change and collaboration."

**Making a difference.** Participants mentioned biology education research allowed them to make a difference in the world (Fig 1), which aligned with their personal values. Specifically, participants cited making a difference as (1) a reason for changing careers from disciplinary science to biology education research, (2) a key alignment between biology education research

and personal values, and (3) a reason for pursuing biology education research. Of the participants who mentioned making a difference, some highlighted the *direct applicability* of biology education research to the real world. For example, Caden expressed he can directly apply his work to the classrooms and students he teaches, allowing him to see the real-world impacts himself.

> I think part of it is the application part, right? For us, a lot of times we collect data, and then we get to go and act on that research, we get to see how it's applied in real-world settings and in the classroom. I think that sets what we do as educational researchers apart, you know, in a setting where we also teach, apart from maybe some bench or field researchers that don't really get to see the application of their work.

Similarly, a survey participant mentioned the applicability and immediate positive difference biology education research has on their teaching, their colleagues' teaching, and departmental-level decisions.

> I didn't feel like my disciplinary science was being applied, and I see the immediate applications of discipline-based education research to my teaching, my colleagues' teaching, and department-level decisions. I feel like I'm making a positive difference that I can see and apply, whereas I wasn't seeing the applicability of my disciplinary research.

Other participants more generally spoke about making real world impacts. For example, August mentioned they switched from disciplinary biology research to biology education research because they didn't feel their disciplinary research was going to have real-world impacts that would help people get their basic needs met, and they felt biology education research would help them directly impact the world.

> It [August's disciplinary research] wasn't going to impact the world, and I was feeling like I needed to do things that had real world impacts. . .that feeling of 'oh, my god, I can't just sit back when people aren't getting their basic needs met.'

August's value of making a difference in the world spurred them to apply to and begin a biology education research postdoctoral fellowship, leaving their disciplinary biology research behind after graduating with a disciplinary biology PhD.

A survey participant similarly noted their disciplinary research mattered less than their biology education research, highlighting the importance of impacting many students through teaching biology and improving biology education through research.

> My disciplinary research was cool and important, but in some sense it didn't matter. . . I wanted to interact with people and do something that directly mattered. I am convinced that biology is important and liberating for students to learn. But if you teach 100 students well, you teach 100 students well. If you do biology education research, you could potentially impact many, many more than just that.

While some participants mentioned biology education research allowed them to make a bigger difference in the world than their disciplinary science research, another participant mentioned both their disciplinary research and their biology education research made a difference in the world, but they simply shifted their focus to helping students. Specifically, Mimi discussed her disciplinary research as a cancer biologist, noting it helped people. However, the

work she conducts now as a biology education researcher also helps people. What's the difference? Her current work helps students whereas her previous work helped people suffering with cancer.

> I mean, but also when I thought about getting a PhD in biology, like in the more traditional sense, I thought, well I'll be a cancer biologist and cancer biology is also a way to help people, so I think like my motivations were always like to help people. I just shifted from wanting to study cancer and help people who have cancer to think of how I can help students. That is my focus now.

The potential to make a difference through biology education research influenced the professional decisions of these participants. Researchers who were not fulfilling this value with their disciplinary science research were driven to shift from their disciplinary science research to biology education research.

Our finding that faculty members joined biology education research to make a difference in the world aligns with previous literature. One study exploring why recent biomedical science PhD graduates left academia to pursue other ventures also cited their desire to make a difference in the world [46]. For example, one participant in that study expressed how they wanted to make a difference through their career, and they could not envision accomplishing this through disciplinary research studying molecules. In our study, August echoed this same sentiment, noting that their loss of interest in their disciplinary science research and their pivot towards biology education research was motivated by needing to make a difference in the world. Despite having the same motivations for leaving disciplinary biology research in academia, the participants in our study pivoted towards biology education research, staying in academia, rather than pursuing nonacademic career pathways as the PhD graduates did in previous literature [46]. Perhaps the difference in choices (i.e., either stay in academia but pursue a different research pathway versus leave academia altogether) is a result of a difference in underlying interests between the two study populations, nonacademic career opportunities specific to biomedical researchers, or a lack of knowledge among participants in the previous literature study about biology education research as a career option [46]. The latter is a code we found in our present study; see discoveries of opportunities.

Another study found STEM students and graduates wanted to make a difference in the world. Specifically, a qualitative study of the education and career goals of 38 black and Latinx STEM, largely undergraduate, students found that students wanted to promote equity through helping their STEM community, helping marginalized ethnic and racial communities, and addressing global disparities; they wanted to make a difference. However, many of them acknowledged they do not perceive STEM to be service-oriented or social justice focused, causing dissatisfaction in the STEM major/field [47].

In biology education research, and discipline-based education research more broadly, there are many national and international initiatives directed at making a difference in STEM education. For example, the United States' National Science Foundation lists "diversity in STEM" and "education and training" as two of their key focus areas, amongst others (e.g., "biology," "chemistry and materials," and "engineering") [48]. These focus areas are linked to making a difference in diversity, equity, and inclusion in STEM fields, and are likely of interest to discipline-based education researchers, including the biology education research participants surveyed and interviewed in the present study.

**Change and collaboration.** Several participants mentioned they were drawn to aspects of the culture of biology education research compared to that of their respective disciplinary science, emphasizing the open-mindedness (i.e., open to change), collaborative nature, and the

inclusive and friendly atmosphere of the biology education research community ([Fig 1]). For example, Jane mentioned the biology education research community is open to a variety of people communicating a myriad of ideas.

> I feel like the discipline-based education research communities are more open to change. I think they're open to hearing and talking about it. I feel like there is more of a community where different people can talk; different people can express themselves. There's more than one journal and more than one listserv.

Similarly, Caden expressed his appreciation for the progressive nature of the biology education research community.

> I've always felt like this group is really about moving forward as a field and helping researchers and the research field grow. I feel like there's a lot of positivity, encouragement, and focus on mentorship and bringing up young researchers in the field. That is something that I value a lot.

Additionally, participants noted biology education research is collaborative. Specifically, August noted they really enjoy collaborations in biology education research especially because biology education research folks are friendly and supportive.

> I really enjoy collaborations and forming relationships with folks and then working with them. I think every project I do with other people is way more fulfilling than projects that I do alone, and they're better because there are more heads. I don't think I've had like one thing that has been consistent through my whole career, except for different collaborations with different folks. It's always exciting, and I learn something, and we put together a better project than I ever could. I make sure those collaborations are more than a working relationship. It's a friendship, it's being there for each other, and it's really making it a full commitment to each other.

From participant recounts, it is clear they were attracted to the open-mindedness and collaborative nature of the biology education research community. Specifically, participants valued its openness to change and to collaboration.

The biology education research literature often reveals the open and collaborative nature of the field. For example, prior literature advocated for interdisciplinary, collaborative research between biology education researchers and biology learning sciences [49], developmental cognitive psychologists [50], and other discipline-based education researchers (e.g., physics education, chemistry education, geosciences education [51] to bring diverse perspectives, viewpoints, experiences, and expertise to bear on a problem, resulting in more and better solutions [49–51]. *CBE-Life Science Education*, a popular biology education research journal, even released a special issue in response to calls to promote research across disciplinary boundaries in STEM education [52]. The biology education research community clearly supports change and collaboration, so the fact that our faculty participants mention their appreciation for this openness to change and collaboration demonstrates a direct alignment between at least one of their values and the biology education research academic culture.

This open and collaborative nature of the biology education research community persists across discipline-based education research fields. For example, engineering education researchers proposed engineering education would be served by collaborations between discipline-based education research (e.g., biology education research) and Cognitive Psychology

communities [53]. Similarly, engineering education researchers have promoted collaborations across STEM discipline-based education research fields (e.g., biology education, physics education, chemistry education, geosciences education), outlining the benefits to the science and engineering education communities [54]. Ultimately, this alignment represents one of the three social influence processes that impact career motivations (i.e., values), according to the Social Influence Model [22].

### Personal interests

According to Social Cognitive Career Theory, personal interests influence career choices and trajectories [23]. Our findings reflected this with three codes: for the love of teaching, evidence-based approaches to education, and the best of both worlds (Fig 1). These three personal interests are closely related. As such, after presenting each code, we discuss the three codes together.

**For the love of teaching.** Faculty member participants acknowledged their love of teaching as part of or as the entire reason they conducted biology education research (Fig 1). For example, Wanda noted that in graduate school she was interested in the disciplinary biology research she was conducting, but she was *really* interested in teaching. This interest prompted her to participate in extra teaching assistantship assignments, driving to other universities to teach while in graduate school. After graduating with her PhD, she pursued a teaching postdoctoral position and realized she wanted to conduct biology education research. Subsequently, she applied for and obtained a postdoctoral researcher position in biology education research.

Similarly, August mentioned they were a mentor to teaching assistants and a graduate trainee of the Teacher Training program in graduate school, causing them to take interest in a postdoctoral position in biology education research. However, they thought the position would have them "teaching and doing some research." They did not understand they would not be doing any teaching, the original interest that motivated them to take the postdoctoral position. However, they noted, "Yeah, I took the position and still didn't totally understand what it was, but it all worked out wonderfully."

Caden also pursued education research due to his interest in teaching. Specifically, he was already teaching biology courses as a lecturer after he received a master's degree in disciplinary biology. During his position as a lecturer, he sought out training to improve his teaching. Specifically, he noted:

> I experienced some severe pedagogical discontentment with what I was doing in the classroom, so I began to seek out ways to learn more and to gain some professional development in teaching. I wasn't as geared towards getting in the research side of things, but I was really motivated by trying to be a better instructor. I ended up signing up and taking some graduate level science education courses at the local university. I was taking like one or two classes a night just to become a better instructor.

This experience of taking night classes in science education landed him a job offer for a research assistant position in one of his professor's labs conducting science education research. This ultimately put him on the path to conducting biology education research later in his career trajectory.

Although many participants discovered their love of teaching during graduate school, Mimi always wanted to be a college instructor and knew about her love of teaching before

graduate school. In fact, she went to graduate school to get a PhD in biology, so she could teach biology.

> I always wanted to go into teaching, and my family members were teachers. I wanted to go into education, and biology was just a thing I was really good at, and I liked it. I went to undergrad and fell in love with the college atmosphere and thought this would be really fun. Like, I really want to work with college students and help shape their careers and all that. Beforehand, I wasn't really sure if I wanted to do K-12. But once I was in college, I was like, okay, this seems like it would be a fun gig. I was told that if you want to be a university professor, then you have to get a PhD. So, I thought, I will get a PhD, and I was the first one in my family to go to grad school and to do all that.

Commonly, the love of teaching prompted participants to seek positions that aligned with this interest, leading them, in this case, to pursue a career in which they conduct biology education research.

**Using evidence-based approaches to education.** Participants were personally interested in evidence-based approaches to education. We defined evidence-based approaches as those responses that described being interested in doing research that improved participants' teaching or improved education more broadly (Fig 1). For example, one survey participant explained they were motivated to pursue biology education research after learning they could conduct biology education research to answer their questions about how to improve biology teaching and learning.

> I began to see that there was a way to conduct educational research and answer questions that I had about teaching and learning. Eventually, with the encouragement of several faculty mentors, I decided to pursue a Ph.D. I wanted to know how to conduct high quality biology education research, so I found a program in Science Education where I could pursue an emphasis in college science education.

Another survey participant explained their reasoning for changing careers from disciplinary science to pursue biology education research, citing conducting research to provide evidence-based approaches to support other instructors.

> I got more and more excited about the possibility of actually conducting the research and supporting other instructors. . . which made my transition from scientist to science educator, to science education researcher, complete.

Another survey participant explained their motivation for pursuing biology education research centered on collecting evidence to develop evidence-based approaches proven to help teach better, reach more students, and help more students identify as scientists.

> I felt my strengths lay not in doing ecology research, but in teaching ecology (and biology more generally) using evidence-based approaches. That there is a need for research to identify those approaches fuels my research on a daily basis. How can we teach better, reach more students, and help more individuals see themselves in science? That's what drives me.

Finally, a survey participant explained their motivation for pursuing biology education research was rooted in measuring the effect of curriculum changes on student outcomes.

I was interested in curriculum development, as it is the intersection between disciplinary science and education. Then, I became interested in measuring student outcomes.

Commonly, an interest in evidence-based approaches to education prompted participants to seek positions that aligned with this interest, making the pursuit of a career in which they use evidence to conduct biology education research a natural fit.

**Best of both worlds.** Two of the participants' main interests, the love of teaching and evidence-based approaches to education, commonly appeared together, resulting in a third popular code: the best of both worlds (Fig 1). For example, a survey participant explained that biology education research gave them a way to combine their love of teaching and research.

. . . gave me a way to combine teaching and research and that's something I was able to get excited about!

Four other survey participants echoed an appreciation for the same sentiment.

I always loved teaching, and this is a way to navigate two fields (science and education). I get to wear multiple hats and study what I love.

It's hard to explain, but the field was perfect for me. It let me merge my love of teaching with research in ways that positively impact learners and learning.

I am encouraged by my science background to apply the scientific method and empirical approaches to the classroom. It blows my mind how many teaching statements I've read by top-notch researchers that do not have a single reference in them.

I am passionate about teaching and wanted a career with more teaching, and I wanted to be evidenced-based in my teaching.

Together, an interest in teaching and evidence-based approaches to education prompted participants to seek positions that aligned with these interests. These two strands of scholarship are inherent to biology education research.

**Personal interests (love of teaching + evidence-based practices + best of both worlds.** According to Social Cognitive Career Theory, personal interests influence career choices and trajectories [23]. Participants frequently mentioned their interest in teaching, evidence-based approaches to teaching, or an interest in both (i.e., the best of both worlds) as the main motivators to conduct biology education research. Due to the overlap in codes, we discuss these codes together.

Biology education research emphasizes the importance of teaching and research combined (the best of both worlds), often through the investigation and use of evidence-based teaching practices to support student learning [6]. Prior literature documented this combination of teaching and research, showing the majority of Science Faculty with Education Specialties (SFES) throughout 23 campuses in the California State University system reported spending their time teaching, soliciting external grant funding, and publishing peer-reviewed articles [9].

The majority of tenure track research faculty in the United States are required to teach within their discipline, and the majority of teaching faculty completed advanced degrees that required they conduct research. Given the professional proximity of these activities, the finding that many of our survey and interview participants applied skills from one domain to the other was not surprising. In fact, Vision and Change in Biology Education, a document that outlines the fundamental concepts to guide how we teach [6], suggests that students with a

biology degree should be able to apply quantitative reasoning and the nature of science across interdisciplinary contexts (such as teaching). Further, scientists are trained to think critically about how and why they choose approaches to addressing fundamental questions in their work. This has led some to apply a critical lens and the scientific method to teaching as well [55, 56]. While we would expect most or all trained scientists to iteratively improve their teaching based on data, this is not what national studies suggest [57]. Rather, broad patterns of observations of teaching suggest STEM instructors continue to teach using undisrupted lecture rather than use evidence-based active learning strategies [57]. In the current study, we found a central motivation to pursue biology education research was the ability to combine research and teaching in the context of a biology department. The importance of effective instruction to enhance student outcomes in biology and STEM and the urgent need to teach an ever-expanding pool of incoming students may have increased professional opportunities for individuals who possess a love of teaching and an interest in evidence-based practices. Based on our surveys and interviews, we suggest those academic roles selected for faculty who wish to combine quality teaching and research.

## Contextual factors

According to Social Cognitive Career Theory, career choices and trajectories are influenced by contextual factors [23]. Our findings reflected this with five contextual factor codes: discoveries of opportunities, the role of literature, mentorship, experiences with sexism, and economic drivers (Fig 1). Ultimately, some of these contextual factor codes acted as barriers to career development while others acted as an impetus to begin conducting biology education research.

**Discoveries of opportunities.** Participants mentioned that, prior to pursuing biology education research, they were unaware of biology education research or careers in education research. Participants also mentioned that salient events or actions led to a career in biology education research by happenstance. These responses highlight how discoveries of opportunities in biology education research altered career trajectories and how lacking awareness about these opportunities served as a barrier to careers in biology education research. Multiple survey responses echoed a previous lack of awareness concerning biology education research. Two examples follow:

> I wasn't aware of biology education research as a legit pursuit, and originally thought I needed to be "just an instructor" and do biology education research (but not by that name) on the side. So, I went to some conferences and wrote some papers and then started realizing I could make a legit research career out of biology education research. So, I applied for a tenure-track faculty position in my mid-40s, got it, and there you go.

> I didn't know there was a field of research in curriculum and instruction—once I knew the field existed, I knew that is what I wanted to do my doctoral training in.

While the participants' values and interests remained the same, the participants only pursued careers in biology education research after discovering opportunities within the field. At the time that our participants were entering the field of biology education, it was an opportunistic niche limited by the fact that many people, even those interested in teaching and evidence-based approaches, were not aware of it.

In other words, a career in biology education research did not result from careful planning but rather emerged after discoveries of opportunities in biology education research. This lack of a clearly defined career goal has previously been demonstrated in the literature. Specifically, one study found that out of 38 recently graduated biomedical PhD students, more than half

(55.2%) of them described entering their PhD program without clearly defined career goals [46]. Research examining whether science faculty with education specialties (SFES) were hired for their SFES job or transitioned into the role found that (47%) of the participating SFES transitioned into their education research roles rather than initially seeking them out and being hired for that role [7].

Our finding that the participants in our sample only pursued biology education research positions after first receiving graduate training in disciplinary science highlights two key problems. The first key problem is that participants did not know about biology education research as a career pathway at the outset of their education. This lack of knowledge acts as a gatekeeper to entering biology education research. The second key problem is that graduate disciplinary education research training also acts as a gatekeeper to the field of biology education research, especially when considering the many negative contextual factors the participants faced during their graduate disciplinary biology research (e.g., negative experiences with mentors and sexism), which we discuss in a later section (see contextual factors). Aside from the negative factors faced by the participants in our study during graduate school, literature in postsecondary education demonstrates undergraduate students often switch majors from STEM to other majors due to problems with STEM instructor pedagogy, a concern cited by 90% of students who switched to non-STEM majors and by 74% of students who persisted in STEM [58]. Students who experienced firsthand problems with STEM instructor pedagogy may be interested in contributing to biology education research but may not progress along STEM career pathways and towards biology education research. Furthermore, students who identify as persons excluded due to their ethnicity or race (PEERs) leave biology at higher rates than non-PEERs [59], additionally illustrating problems with the common pathway from a graduate degree in biology to a career in biology education research.

In the time since the faculty participants in this study were graduate students in biology, opportunities in biology education research, and discipline-based education research more broadly, have become more abundant and clearly advertised. For example, a handful of institutions have developed more clear pathways for pursuing Biology Education Research (e.g., University of Northern Colorado's PhD in Biological Education; https://www.unco.edu/nhs/biology/). Additionally, in the 2023 fiscal year alone, the National Science Foundation awarded 14% of their budget to STEM education research [48]. Portions of this funding include awards for Building Capacities in STEM Education (BCSER), a program that supports the development of individuals to conduct quality STEM education research that "will enhance the nation's STEM education enterprise and broaden the pool of researchers that can conduct fundamental research in STEM learning and learning environments, broadening participation in STEM fields, and STEM workforce development" [60]. With increasing funding opportunities in discipline-based education research, more people are likely to become aware of the viability of this research field.

**The role of literature.**   Multiple interviewed participants cited biology education research literature as an important part of their journey to biology education research. For example, Wanda recalled reading high-impact science publications weekly during graduate school, causing her to come across the first discipline-based education research article she read. Specifically, Wanda stated, "I read *Science* and *Nature* every week, and I came across Carl Weiman's work in graduate school." Notably, Carl Weiman is an American physicist and Nobel laureate, who presently conducts physics education research. Similarly, Martha explained her first exposure to biology education research was through literature. She mentioned her predecessor left behind Biochemistry and Molecular Biology Education (BaMBEd) journals. After Martha noticed her students were not understanding a particular lecture, she picked up one of those BaMBEd copies and started reading. She found the research helpful and applicable to

her teaching. Caden also mentioned he learned about biology education research through research articles.

> I was teaching at a community college, and I had never been taught to teach. A colleague of mine had a bunch of American Biology Teacher Journals. . . That's the first time I saw actual research on biology education.

Aside from discovering biology education research in biology education journals, some participants learned about biology education research from calls to action. This was the case for Jane, when she read "Rising above the gathering storm: Energizing and employing America for a brighter economic future" [61].

> After my postdoc, I did an [industry job]. I think that's where I first kind of started learning about it [biology education research]. It was right after the Gathering Storm report had come out from the academies, so that's probably how I first came across it [biology education research] . . ., realizing for the first time that people were studying this as like a disciplinary science and not just as like a hobby or a policy kind of thing.

Biology education research articles and reports inspired many of the participants in our study, seeding the possibility of a different path that eventually influenced their career trajectory.

Participants mentioned biology education research journals were a source of inspiration to conduct biology education research. Usually, literature acted as a catalyst to conduct biology education research because the participants were previously unaware it was a field of research. Participants specifically mentioned reading biology education research in high-impact journals, such as Science (https://www.science.org/). Clearly, other high impact journals, such as Nature (https://www.nature.com), Proceedings of the National Academy of Sciences (PNAS; https://www.pnas.org/), and BioScience (https://academic.oup.com), can elevate biology education research, allowing it to reach new research fields and the people who populate them. Similarly, open-access journals increase opportunities for non-subscribers to read biology education research. Examples of open-access journals that publish biology education research include CBE- Life Sciences Education (https://www.lifescied.org), Journal of Microbiology and Biology Education (https://journals.asm.org), and the International Journal of STEM Education (https://stemeducationjournal.springeropen.com/). Overall, access to journals that publish biology education research was many times critical for the participants in our study, allowing them to discover the field of biology education research and subsequently pursue such research in their career.

**Mentorship.**   Both positive and negative experiences with mentors influenced the participants' decisions to pursue a career in biology education research. Notably, participants often had a negative experience with disciplinary biology advisors and a positive experience with biology education research mentors. For example, Wanda mentioned her biology education postdoctoral advisor "was personally very supportive. . .always very open-minded, and whatever people said they wanted to do, she would help them get that job, so that was very encouraging." This was a more supportive environment than the one she had during her disciplinary biology PhD experience. Specifically, Wanda noted her PhD advisor "clearly didn't think this education stuff was worthwhile," despite him being a "good guy" and letting her do her "teaching and stuff."

Other participants had more discouraging experiences. Joan explained she was interested in taking biology education research courses during her disciplinary biology PhD, but her advisor did not support this interest.

I wanted to take an education class. I thought that would be super fun. I was really discouraged from doing that and was just told, "That's not what we're doing. You need to spend more time in the lab. You know, I gave you time off to do your preliminary exams." It was just super toxic. I mean, the gaslighting was insane. A week later, he [her advisor] found out that his funding got cut. Two graduate students in his lab came from the same funding source, and he had to cut someone. I was the one that got cut and honestly that was the best thing that ever happened to me because I think I would have just driven off a bridge otherwise. I mean, I really was just miserable.

Aside from not supporting her interests, her disciplinary biology advisor cut Joan from the lab shortly after discussing her interest in biology education. However, she noted her previous advisor did help her find a new advisor who had a positive impact on Joan's life.

He [her original advisor] helped me find a new advisor, and my new advisor had a dual appointment in education and in biology. She did STEM education stuff, and she pretty much just took me under her wing. I had no education research experience whatsoever, and, our first semester working together, we published a paper. I mean it was just a much better working environment, and she was a great advisor. I was her first grad student, so she was really new. We learned a lot together, which I think was really fun. She introduced me to education research, and I started taking all these education classes: qualitative analyses, inclusive education, and multicultural education. It kind of just all worked out.

Joan was not the only participant who was removed from a disciplinary biology lab. Martha detailed her experience with her first research technician position mentor, suggesting he wanted to get rid of her despite having unlimited funding. This experience caused her to lose faith in herself.

He technically had funding that couldn't run out, but if you want to get rid of somebody. . .and he was notorious for doing that. . .so he dropped me two days before the end of my six months where they have to keep you on, but I was picked up almost immediately by the woman that I did my master's degree with, who spent a lot of time repairing my self-efficacy and confidence in myself, which had been totally destroyed by you know, a guy in an office that said, "Oh, don't worry. Everyone who's ever worked for me has cried in my office."

Participant experiences with mentors obviously left their mark, with mentors from biology education research often having a positive impact on participants and pulling them further into biology education research and mentors from disciplinary biology research either having a neutral or negative impact on participants and pushing them away from disciplinary biology research.

Mentors were a key contextual factor in the participants' pursuit of biology education research and often a key factor in their dissatisfaction with the climate of their disciplinary science research. From previous literature, we know mentoring relationships while students are in graduate school can have positive and negative effects [62–69]. For example, in a study that interviewed 31 STEM postdoctoral researchers at a large, private research-intensive university in the southeast United States about their relationships with formal and informal mentors and how those experiences influenced their attitudes toward academia, finding that the postdoctoral researchers considered alternate career paths because of their STEM advisors' work-centric lifestyle, differences in values, and difference in personalities [65]. However, when the

postdoctoral researchers and their advisors had similar lifestyles, values, and personalities, then the postdoctoral researchers were encouraged to stay in STEM and seek out positions like their advisors' position [65]. Similarly, another study found that matching personalities contributed strongly to mentees' positive mentoring experiences [63]. However, positive advisor-advisee relationships in disciplinary biology research were not the case for many of the interviewed faculty participants in our present study. Often, toxic environments led to participants leaving their disciplinary biology research and pivoting to biology education research because it more closely aligned with their values and interests.

**Experienced sexism in disciplinary science.**   Some of the interviewed and surveyed participants encountered sexism, discrimination, and marginalization based on their gender during their disciplinary biology research. One survey participant recounted an experience with her advisor at the time, right after she had a baby and just as she started a PhD program. She explained her advisor was not supportive of her leaving the lab in the early afternoon to care for her newborn and eventually gave her an ultimatum to either put her child in daycare or quit the lab. She chose to leave the lab, and, when she told her advisor of her decision, he reprimanded her, citing her gender, motherhood, and religious views as reasons he knew she would not succeed.

> I had a baby at the end of my master's degree, so I started my PhD with a 4-month-old. I was doing my best to fulfill all of my responsibilities, arriving in the lab at 6:00 a.m. but leaving by early afternoon, so I could swap my husband with the baby. My advisor, who had kids of his own and who I thought would be flexible with improvised schedules, was not at all supportive and told me I needed to put my kid in daycare and be in the lab from 10:00 a.m. to 6:00 p.m. or get out. So, I got out. Just as an aside, when I left the lab, my advisor yelled at me that he knew I wouldn't succeed because I was a Woman, a mother, and a [particular religious group]. Needless to say, he was a misogynist. But it almost ended my career entirely.

Another surveyed participant mentioned facing sexist experiences when she was studying biology. She did not get into specifics but mentioned she experienced sexism on multiple occasions.

> Going from being a minority in a field [a sub-field of biology] to a majority member of a field [discipline-based education research] has its perks. I had a lot of sexist experiences in [sub-field of biology], and I worked with almost only men. Now I work with nearly only women. It's wonderful.

As a final example of experiencing harassment due to their gender, a survey participant admitted they were commonly harassed due to their gender, and this caused them to leave their disciplinary science.

> I experienced a TON of harassment during my PhD field work, and in retrospect I can see that was a clear signal that I did not belong in the discipline. But I did like teaching, so. . .

She alluded to joining biology education research because she liked teaching, and she did not like the harassment she endured during her doctoral field work in disciplinary biology research.

Together, these experiences with sexism, a negative contextual factor, represent a push away from disciplinary biology research. In any disciplinary field, it is a desirable outcome for all to work without the disadvantage of sexism or other forms of discrimination.

Participants encountered sexism during their disciplinary biology career path. According to the literature, experiences of enduring sexist ideology from colleagues are not uncommon [70, 71]. For example, Rita Colwell, a professor at the University of Maryland at College Park and at Johns Hopkins University Bloomberg School of Public Health and former director of the National Science Foundation from 1998 to 2004, wrote a book about her experiences with sexism in science [71]. In a post adapted from her book, she detailed enduring similar sexist ideology and commentary described by some of the participants in our study [70].

While different from outright sexist statements, there are also structural forms of sexism. One structural form of sexism caused one of our participants to leave her disciplinary biology field. One survey participant started a PhD program quickly after giving birth to a baby, changing her daily schedule to outside of her advisor's expectations. Her advisor told her she needed to adopt his schedule or leave the lab. This structural form of sexism affects many women scientists. In fact, nearly one-half of new mothers leave full-time STEM employment after having children [72]. Compare this to nearly one-quarter of new fathers making the same decision. These experiences with outright sexism and structural sexism acted as a push away from disciplinary biology research for participants in our study, a contextual factor they did not mention facing in biology education research.

It is important to acknowledge that in addition to asking about experiences with sexism, we also asked about experiences with racism. Specifically, to elicit instructor experiences of sexism and/or racism, we asked both on the survey and in the interviews: "Do you feel your gender and/or race/ethnicity had anything to do with your decision to leave a disciplinary science and enter biology education research? Why or why not?" Notably, we do not have a section on participant experiences with racism. This is because only one of the participants mentioned a racialized experience. See considerations section for more on this.

**Economic drivers.**   Economically driven contextual factors (e.g., higher pay, job security and stability) impacted several interviewed participants' career trajectories, influencing their choice to conduct biology education research as it is a higher paying and more stable career than non-tenured track lecturer positions. For example, Caden noted the university he was teaching at was cutting their budget. His department chair explained that Caden would likely be one of the first people to lose his job, given he was not hired as a tenure-track faculty member. This caused Caden to look for other jobs that were tenure-track, research-based jobs in biology education. He detailed:

> My department chair called me into his office and said, "[redacted] university's system is really struggling with money right now. We may get cuts to our budget. You are the newest person here and you are non-tenure-track. You would be the first person to go." I'm glad he was super honest with me. I started looking for jobs at the time. I guess two things led me to biology education research which are 1) I realized the job security that comes with a tenure-track position is pretty nice, having almost been fired just because of money issues, and 2) as I was job searching, I actually found this job here at [university], and I was like wow this is what I want to do, like this is the job that I wanted.

In this case, economic drivers, such as budget cuts, led Caden to search for a more permanent job, leading him to apply for and secure a tenure-track assistant professor position in which he conducts biology education research.

In another case, a participant mentioned a higher salary drove them to conduct biology education research rather than focus only on teaching. Martha detailed her frustrating experience with a previous teaching job. Her department chair encouraged all lecturers to apply for

education grants, in addition to keeping up their high teaching load, without offering raises to their salaries.

> The tipping point was receiving an email from the department chair that said, "hey, it looks like people are getting money for education. Why don't you write some grants and bring some money in?" It was sent to all of the non-research instructors that had an 80% to 100% teaching load. I'm like, I'm not going to do all of the work of a professor and make less than half the salary. Are you kidding?

Joan mentioned she went to graduate school to get a PhD, so she would have job security as a tenured professor and therefore could help support her family financially.

> I think a lot of my motivation was, as a professor, I would have money and job security. I grew up dirt poor. No one in my family has any money, so I've pretty much been supporting my family since I was a graduate student. I just thought having a PhD would give me job security.

According to Social Cognitive Career Theory, *personal interests* lead people to pursue a particular training path, however, this path is impacted by *contextual factors*. Here, the real *contextual factors* of salary and job security impacted participants' career paths, ultimately becoming biology education researchers.

Economic drivers such as salary and job security piqued participants' interest in considering tenure-track biology education research positions. Ultimately, pursuing tenure-track faculty positions allowed participants to pursue their interests and values while simultaneously obtaining job security as a tenure-track professor and an increased salary compared to a lecturer position.

Lecturers—a term inclusive of instructors hired to teach one or two courses for a semester, experts or practitioners who are brought in to share their field experience, adjuncts who are generally compensated on a per-course or hourly basis, as well as full-time non-tenure-track faculty who receive a salary—receive substantially less pay than tenure-track faculty positions [73]. In fact, non-tenure track lecturer salaries range from an average of $14,400/year at Wofford College in South Carolina to an average of $54,300/year at Sewanee: The University of The South in Tennessee [74]. Additionally, adjunct faculty members receive an average of $2,979 in public associate's institutions without ranks and an average of $5,557 in public doctoral institutions per course [75]. In comparison, assistant professor salaries range from an average of $33,600/year at Universidad Politecnica De Puerto Rico in Puerto Rico to an average of $137,000/year at Stanford University in California [74]. This demonstrates tenure-track faculty are paid more, on average, than lecturers. An analysis of annual salaries of 700 full-time lecturers and tenure-track faculty at 37 public Ph.D.-granting departments of economics likewise found a similar pay discrepancy between lecturers and tenure-track faculty [76].

In addition to being paid less on average than their tenure-track counterparts, lecturers experience less job security, as the lecturer position requires contract renewal. Roughly three out of every five faculty members (61.5%) were on contingent (i.e., non-tenure track) appointments during the fall 2020 semester. Given that faculty tenure is the only protection for academic freedom in teaching, research, and service, this suggests most lecturers have less job security than their tenure-track counterparts [75]. While the lecturers we interviewed may also have found higher paid positions in disciplinary biology tenure track positions, their extensive teaching experiences made them more fit to pursue research in teaching. Together, the appeal of higher salaries and tenure attracted some of our participants to seek biology education

research as a career, since lecturer positions paid less and offered less job security than a tenure-track position in biology education research.

## Considerations

The authors want to make readers aware of several considerations in interpreting and understanding our research. First, we did not thoroughly explore all aspects of Social Cognitive Career Theory or the Social Influence Model, though this was an intentional decision made by the authors. Specifically, we did not include prompts or questions related to the participants' self-efficacy attitudes in our survey and interviews. However, none of the participants mentioned their self-efficacy, either directly or indirectly, as a motivator for either leaving their disciplinary science or pivoting towards biology education research. Additionally, demonstrated self-efficacy has previously been shown to poorly predict intentions to pursue a scientific career when participants already identified as scientists and internalized scientific values [22]. For this reason, we focused more on the (1) values of both the participants and the values the participants believed the disciplinary sciences and the biology education research community held, (2) the participants' interests, and (3) the contextual factors the participants faced on their career path. However, we acknowledge that an exploration of self-efficacy could have revealed interesting results.

Second, our research participants self-identified as teaching or faculty members that conduct biology education research. While biology education research is its own field, it can be housed under a larger umbrella called science faculty with education specialties (SFES) [8]. With this in mind, we cannot rule out the idea that some of the participants may identify more with the SFES categorization. This limited our ability to discreetly bin each participant as a faculty member who solely conducts biology education research. We acknowledge that some of the participants in our study are multidisciplinary and interdisciplinary, conducting science education research and disciplinary biology research.

Third, participant demographics were largely homogenous, with most participants identifying as white women. We did not report gender or race demographics more clearly than this to avoid participant identification, but we did present findings on how some of the participants' identities influenced their experiences (e.g., some women participants encountered sexism). However, even though we additionally asked about racialized experiences in both our surveys and interviews—because there was only one reported experience with racism, and this experience involved several specific identities—we could not present this finding due to the potential loss of anonymity for the participant. Here, we ask the reader to consider that just because we do not present evidence of racialized experiences in participant's career paths in science, does not mean we did not find any evidence of this. In fact, prior literature documents myriad racialized experiences in STEM [77–80].

Fourth, in the exploration of career pathways, we recognize career decision-making is an ongoing process. However, in this study, we did not explore the faculty participants' satisfaction with their roles conducting biology education research, their current employment, nor the support they received from their institutions. Given previous literature exploring SFES job satisfaction, less than ten percent of SFES felt their academic institutions supported an infrastructure for undergraduate and graduate courses and degrees for science education in the same way it supported an infrastructure for disciplinary science [7]. Additionally, almost 40% of the 59 SFES they surveyed were considering quitting their jobs, feeling a lack of institutional value for their science education efforts, and being overworked and burned out [7]. This suggests that if we had inquired about their current satisfaction with their positions, the biology education research faculty participants in our study may have had some desired improvements

to mention. Work exploring biology education researchers' job satisfaction will be important to conduct in the future to support faculty who are focused on science education.

Fifth, both the present research and most of the reviewed literature on career pathways was conducted prior to a major contextual factor: the COVID-19 pandemic. This is an important consideration given prior literature documented that chemistry postdocs and graduate students experienced serious decreases in research progress and increases in anxiety levels [81] and undergraduate biology students faced barriers to learning during the pandemic including loss of campus resources and difficulty focusing on learning [82]. To our knowledge, specifically how the pandemic and the barriers it presented to STEM students, graduates, and faculty have yet to be empirically evaluated, so we do not yet know how the pandemic has or has not impacted their career paths in STEM. It may be that the pandemic would come up as an important contextual factor that informed STEM career decision-making, if we were to conduct this study in the future.

Finally, we recruited participants from the Society for the Advancement of Biology Education Research (SABER) listserv and the authors' Twitter followers. Notably, while SABER is the largest of its kind in the world, it is based in the United States and its members are largely North American. We acknowledge this study may not be representative of career decisions in biology education research in other places (e.g., European Researchers in Didactics of Biology (ERIDOB); Biosciences Education Australia Network (BEAN)). Additionally, by not asking that individuals forward our invitation to their biology education researcher peers, we may have missed the opportunity to survey those individuals.

## Conclusions

Together, our findings showed our faculty participants came into biology education research due to their values and personal interests, but were additionally swayed by contextual factors (e.g., an unsupportive or supportive disciplinary biology advisor, the ultimate discovery of biology education research literature, fiscal and job security concerns, and sexist experiences in disciplinary science research). These contextual factors acted as either a push from disciplinary biology research or a pull towards biology education research. As a community, we must consider the influence of these contextual factors to make more direct paths towards biology education research accessible for a broader population. For example, we must continue to push to increase our visibility as a viable academic pathway for students in science, because, as it stands, the current path to biology education research seems to be based on exposure to biology education research literature and word of mouth. To reach a wider audience, we support:

1. Publishing in open-access journals such as CBE-Life Sciences Education, Journal of Microbiology and Biology Education, and International Journal of STEM Education, as well as high-impact journals that attract a broad readership.

2. Expanding the roles of professional societies and networks beyond sharing research findings (e.g., The Society for the Advancement of Biology Education Research, The American Society for Cell Biology) or focusing on implementing evidence-based teaching practices in the classroom (e.g., Partnership for Undergraduate Life Sciences Education). This expansion of roles could include sharing educational research training and career path information with society/network members. Additionally, adding reflection concerning who this information is reaching will be important, so societies and networks can continue to reflect and improve their practices around diversity, equity, and inclusion.

3. Biology education researchers inviting other biology education researchers to seminar series at their home institutions, especially for those hosted by biology departments.

4. Disciplinary biology researchers inviting biology education researchers to seminar series at their home institutions.

5. Courses teaching undergraduate and graduate students how to conduct biology education research. Such courses not only widen the biology education research audience, they also can result in biology education research publications (e.g., [83]) and increase student science identity and emotional ownership of research [84].

6. Disciplinary biology researchers in faculty positions supporting their student's research interests, whether they are disciplinary biology focused or biology education research focused. There are many successful examples of careers that combine both biology education research and disciplinary biology research.

7. Biology departments and faculty members supporting their disciplinary biology graduate students' potential desire to add an education component to their education and research.

Additionally, we must continue to advocate for individuals who are opting out of their discipline (be it in biology or elsewhere) due to experiences of discrimination and toxic environments.

Ultimately, these findings illuminate the complex paths towards individual career choices, and our results suggest that to attract a broad and effective academic workforce, we must attend to individuals' values and personal interests and foster positive working environments.

## Supporting information

**S1 Table. Complete open-ended survey questions provided to participants via the online survey software Qualtrics.** These questions asked participants about their focus and experiences with research during their postsecondary education, inquiring particularly about their motivations and interests towards biology education research. Questions written in blue were the focus of our analysis.
(TIF)

**S2 Table. Complete semi-structured interview protocol questions with potential probing questions.**
(TIF)

**S1 Fig. Martha's timeline with descriptive interview excerpts.** Green rectangles or circles represent a professional experience in disciplinary biology research and orange rectangles or circles represent a professional experience in biology education research. White rectangles or circles represent a professional experience where neither disciplinary biology nor biology education research was conducted.
(TIF)

**S2 Fig. Caden's timeline with descriptive interview excerpts.** Green rectangles or circles represent a professional experience in disciplinary biology research and orange rectangles or circles represent a professional experience in biology education research. White rectangles or circles represent a professional experience where neither disciplinary biology nor biology education research was conducted.
(TIF)

**S3 Fig. Jane's timeline with descriptive interview excerpts.** Green rectangles or circles represent a professional experience in disciplinary biology research and orange rectangles or circles represent a professional experience in biology education research. White rectangles or circles represent a professional experience where neither disciplinary biology nor biology education

research was conducted.
(TIF)

**S4 Fig. Joan's timeline with descriptive interview excerpts.** Green rectangles or circles represent a professional experience in disciplinary biology research and orange rectangles or circles represent a professional experience in biology education research.
(TIF)

**S5 Fig. Wanda's timeline with descriptive interview excerpts.** Green rectangles or circles represent a professional experience in disciplinary biology research and orange rectangles or circles represent a professional experience in biology education research. White rectangles or circles represent a professional experience where neither disciplinary biology nor biology education research was conducted.
(TIF)

## Acknowledgments

We would like to thank the biology education researchers who participated in our survey and interviews. We appreciate their time greatly. We would also like to thank Dr. Michael Smith and Dr. Peter Marting for their feedback on the writing of the manuscript and the figure aesthetics. We would like to thank the Ballen lab for critically thinking about our survey and giving us constructive feedback. Finally, we thank Dr. A. Kelly Lane for helpful and careful feedback on the qualitative methods of this article.

## Author Contributions

**Conceptualization:** Emily P. Driessen, Ariel L. Steele.

**Data curation:** Emily P. Driessen, Ariel L. Steele, Robin A. Costello, Peyton Brewer.

**Formal analysis:** Emily P. Driessen, Ariel L. Steele, Robin A. Costello, Peyton Brewer.

**Funding acquisition:** Cissy J. Ballen.

**Investigation:** Emily P. Driessen, Ariel L. Steele, Robin A. Costello.

**Methodology:** Emily P. Driessen, Robin A. Costello, Peyton Brewer.

**Supervision:** Emily P. Driessen, Cissy J. Ballen.

**Visualization:** Emily P. Driessen.

**Writing – original draft:** Emily P. Driessen.

**Writing – review & editing:** Emily P. Driessen, Ariel L. Steele, Robin A. Costello, Peyton Brewer, Cissy J. Ballen.

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
