## [Decision Letter · Decision Letter 0]

24 Jul 2024

PONE-D-24-22006“It let me merge my love of teaching with research”: A qualitative investigation of the career pathways of biology education researchersPLOS ONE

Dear Dr. Driessen,

Thank you for submitting your manuscript to PLOS ONE. After careful consideration, we feel that it has merit but does not fully meet PLOS ONE’s publication criteria as it currently stands. Therefore, we invite you to submit a revised version of the manuscript that addresses the points raised during the review process.

**Dear Authors, your work was reviewed and the final outcome is minor revisions.  Please send the revised revision**  for acceptance. Regards==============================

We look forward to receiving your revised manuscript.

Kind regards,

Yolanda Malele-Kolisa, BDS, MPH, MDent, PhD

Academic Editor

PLOS ONE

Journal Requirements:

2. Thank you for stating the following financial disclosure: "We acknowledge the National Science Foundation for providing funding to CJB (NSF-DUE-2011995)." 

3. Please expand the acronym “NSF_DUE” (as indicated in your financial disclosure) so that it states the name of your funders in full.

5. Please ensure that you include a title page within your main document. You should list all authors and all affiliations as per our author instructions and clearly indicate the corresponding author.

Reviewers' comments:

Reviewer's Responses to Questions

**Comments to the Author**

1. Is the manuscript technically sound, and do the data support the conclusions?

Reviewer #1: Yes

Reviewer #2: Yes

2. Has the statistical analysis been performed appropriately and rigorously? 

Reviewer #1: Yes

Reviewer #2: Yes

3. Have the authors made all data underlying the findings in their manuscript fully available?

Reviewer #1: Yes

Reviewer #2: Yes

4. Is the manuscript presented in an intelligible fashion and written in standard English?

Reviewer #1: Yes

Reviewer #2: Yes

5. Review Comments to the Author

Reviewer #1: Summary

This study surveyed and interviewed discipline-based education researchers about their career trajectories. The values and interests, role of mentorship, and factors influencing their career trajectories provide valuable insight into DBER careers. The research team used the Social Influence Model and Social Cognitive Career Theory to develop a survey and interview questions to guide their work. The work draws from two frameworks and uses them to advance knowledge of the career influences of DBER scholars. The values and interests identified and contextual factors help support current and future DBER scholars.

The survey instrument was revised based on expert feedback, and the research team focused on establishing trustworthiness throughout. Thus, the researchers carefully interpreted survey data and invited members for follow up interviews. The methods used by the research team not only ensured the privacy and consent of participants, but also were designed helped elicit honest responses.

Lines 230-234: Did patterns or themes emerge from the research described by survey responses and their open-ended career trajectory responses?

Lines 450-478: In addition to the Gibbs and Griffin 2013 results, are there more recent survey findings from the NIH and National Academies? It would be useful to contrast the 2013 results with opinions during/after the pandemic.

Line 1115: This could be an opportunity to change the role of professional societies and networks (SABER, PULSE, ASCB) to share educational research training and career paths with their members to reflect and improve.

Reviewer #2: This is very important research in an under-researched area. Understanding the values, personal interests, and contextual factors that influence career paths in biology education research can help improve the field.

The authors must address the following:

• The study design was not stated.

• It is unclear if this is a mixed methods study; the authors mention open-ended surveys and follow-up interviews.

• While the recruitment methods are mentioned, the sampling technique is not clear.

• It needs clarification whether the authors mean they did not collect any demographic information for participants when they say, “elected to omit demographic information on these participants to decrease chances of participant identification.” Are there any other participant characteristics collected to describe the study sample?

• Who conducted the interviews, and what are their credentials and experience?

• The authors did not mention obtaining consent for recording the Zoom interviews.

• Data saturation was not addressed.

6. PLOS authors have the option to publish the peer review history of their article (what does this mean?). If published, this will include your full peer review and any attached files.

Reviewer #1: **Yes: **Carlos C. Goller

Reviewer #2: No

---

## [Author Response · Author response to Decision Letter 0]

13 Sep 2024

PONE-D-24-22006

“It let me merge my love of teaching with research”: A qualitative investigation of the career pathways of biology education researchers

PLOS ONE

Dear Dr. Driessen,

Thank you for submitting your manuscript to PLOS ONE. After careful consideration, we feel that it has merit but does not fully meet PLOS ONE’s publication criteria as it currently stands. Therefore, we invite you to submit a revised version of the manuscript that addresses the points raised during the review process.

Dear Authors, your work was reviewed and the final outcome is minor revisions. Please send the revised revision for acceptance. Regards

We look forward to receiving your revised manuscript.

Kind regards,

Yolanda Malele-Kolisa, BDS, MPH, MDent, PhD

Academic Editor

PLOS ONE

Journal Requirements:

*We reformatted our manuscript to meet PLOS ONE’s style requirements, including those for file naming. 

Thank you for stating the following financial disclosure: "We acknowledge the National Science Foundation for providing funding to CJB (NSF-DUE-2011995)." 

*We added the requested change to our cover letter in the second to last paragraph.

Please expand the acronym “NSF_DUE” (as indicated in your financial disclosure) so that it states the name of your funders in full.

*We added the requested change to our cover letter in the second to last paragraph.

We note that you have indicated that there are restrictions to data sharing for this study. For studies involving human research participant data or other sensitive data, we encourage authors to share de-identified or anonymized data. However, when data cannot be publicly shared for ethical reasons, we allow authors to make their data sets available upon request. For information on unacceptable data access restrictions, please see http://journals.plos.org/plosone/s/data-availability#loc-unacceptable-data-access-restrictions. 

*The datasets generated and analyzed during the current study are not publicly available due to concerns of potential participant identification. Data are available on request to the corresponding author only. We are not allowed to send the interview transcripts and survey results to a third party due to limitations with our institutional review board approval (Auburn IRB protocol no. 21-354 EX 2108).

If there are ethical or legal restrictions on sharing a de-identified data set, please explain them in detail (e.g., data contain potentially identifying or sensitive patient information, data are owned by a third-party organization, etc.) and who has imposed them (e.g., a Research Ethics Committee or Institutional Review Board, etc.). Please also provide contact information for a data access committee, ethics committee, or other institutional body to which data requests may be sent.

*Due to the potentially identifiable information in our interview data we are unable to share a de-identified data set. The following is the contact information for Auburn University Human Research Protection Program. Email: IRBAdmin@auburn.edu; Phone: 334-844-5966.

If there are no restrictions, please upload the minimal anonymized data set necessary to replicate your study findings to a stable, public repository and provide us with the relevant URLs, DOIs, or accession numbers. Please see http://www.bmj.com/content/340/bmj.c181.long for guidelines on how to de-identify and prepare clinical data for publication. For a list of recommended repositories, please see https://journals.plos.org/plosone/s/recommended-repositories. You also have the option of uploading the data as Supporting Information files, but we would recommend depositing data directly to a data repository if possible. Please update your Data Availability statement in the submission form accordingly.

*As mentioned above, we are unable to share a de-identified data set due to potentially identifiable information in our interview data.

Please ensure that you include a title page within your main document. You should list all authors and all affiliations as per our author instructions and clearly indicate the corresponding author.

*We added a title page and - in it - included the requested information. 

*We have thoroughly reviewed our reference list to ensure it is complete and correct. We have not cited any papers that have been retracted. To respond to some reviewer comments, we have added references. We disclose that clearly in our response to reviewers. Additionally, to meet PLOS One’s formatting requirements, we reformatted our references using the Vancouver citation style.

Reviewers' comments:

Reviewer's Responses to Questions

Comments to the Author

1. Is the manuscript technically sound, and do the data support the conclusions?

Reviewer #1: Yes

Reviewer #2: Yes

2. Has the statistical analysis been performed appropriately and rigorously? 

Reviewer #1: Yes

Reviewer #2: Yes

3. Have the authors made all data underlying the findings in their manuscript fully available?

Reviewer #1: Yes

Reviewer #2: Yes

4. Is the manuscript presented in an intelligible fashion and written in standard English?

Reviewer #1: Yes

Reviewer #2: Yes

5. Review Comments to the Author

Reviewer #1: 

Summary

This study surveyed and interviewed discipline-based education researchers about their career trajectories. The values and interests, role of mentorship, and factors influencing their career trajectories provide valuable insight into DBER careers. The research team used the Social Influence Model and Social Cognitive Career Theory to develop a survey and interview questions to guide their work. The work draws from two frameworks and uses them to advance knowledge of the career influences of DBER scholars. The values and interests identified and contextual factors help support current and future DBER scholars.

The survey instrument was revised based on expert feedback, and the research team focused on establishing trustworthiness throughout. Thus, the researchers carefully interpreted survey data and invited members for follow up interviews. The methods used by the research team not only ensured the privacy and consent of participants, but also were designed helped elicit honest responses.

*We appreciate the reviewer’s thoughtful summary of our work, and we address all of their comments and suggestions below.

Lines 230-234: Did patterns or themes emerge from the research described by survey responses and their open-ended career trajectory responses?

*To address this comment, we added clarifying text to better explain how we assessed participant responses. It was not part of our research design to analyze the data for patterns or themes. For example, we did not use thematic analysis. Rather, it was our design to see how the constructs from the Social Cognitive Career Theory and the Social Influence Model (i.e., values, interests, and contextual factors) played out in these participants' career paths and decision-making. With this in mind, we focused on coding the data, which represent small units of information that represent values, interests, or contextual factors. Upon reading the reviewer’s comment we re-read our paper and noticed we used the word themes in our figure to talk about the values, interests, and contextual factors, and we now call these “constructs” both in the figure and in the text, so the word theme does not appear in our paper. Additionally this reviewer comment made us realize we needed to more accurately describe our coding process. This involved removing the Hsieh & Shannon (2005) citation and replacing it with Saldana’s (2015) description of first and second cycle qualitative coding. We now describe how we used in vivo coding for our first cycle of coding and focused coding for our second cycle coding, both for the survey analysis and the interview analysis (Lines 306-397).

Lines 450-478: In addition to the Gibbs and Griffin 2013 results, are there more recent survey findings from the NIH and National Academies? It would be useful to contrast the 2013 results with opinions during/after the pandemic.

*After reviewing the literature to respond to this comment, we found several relevant data sources, but none were exactly what we thought the reviewer was asking for nor did many of them address the code “making a difference” — aside from McGee & Bentley (2017) — which is where we discussed the Gibbs and Griffin (2013) study. With this in mind, we added text and the one relevant reference we found (McGee & Bentley, 2017) (Lines 504-510; 1320-1321). Additionally, we added text and references in the considerations section concerning the impacts of COVID on STEM undergraduate and graduate students and that we do not yet know how these have or have not (likely have) impacted their career paths (Lines 1111 - 1121; 1419-1424). 

New References added:

Driessen, E., Beatty, A., Stokes, A., Wood, S., & Ballen, C. (2020). Learning principles of evolution during a crisis: An exploratory analysis of student barriers one week and one month into the COVID‐19 pandemic. Ecology and evolution, 10(22), 12431-12436.

McGee, E., & Bentley, L. (2017). The equity ethic: Black and Latinx college students reengineering their STEM careers toward justice. American Journal of Education, 124(1), 1-36.

Sifri, R. J., McLoughlin, E. A., Fors, B. P., & Salehi, S. (2022). Differential impact of the COVID-19 pandemic on female graduate students and postdocs in the chemical sciences. Journal of chemical education, 99(10), 3461-3470.

Line 1115: This could be an opportunity to change the role of professional societies and networks (SABER, PULSE, ASCB) to share educational research training and career paths with their members to reflect and improve.

*We totally agree! We added this thought to our considerations section as suggested (Lines 1146 - 1154). 

Reviewer #2: 

This is very important research in an under-researched area. Understanding the values, personal interests, and contextual factors that influence career paths in biology education research can help improve the field.

The authors must address the following:

The study design was not stated.

**To address this comment, we added text to the manuscript to identify our study design more clearly (Lines 132 - 134). Our study is an exploratory qualitative study in which we used qualitative methods (i.e., interviews and open-ended survey questions) to learn about the career motivations of people who pursued biology education research (i.e., biology education research faculty). 

It is unclear if this is a mixed methods study; the authors mention open-ended surveys and follow-up interviews.

*To address this comment, we added text to the manuscript to identify our study design more clearly (Lines 132 - 134). Now it is clear that this is not a mixed-methods study. As stated above, our study is an exploratory qualitative study in which we used qualitative methods (i.e., interviews and open-ended survey questions) to learn about the career motivations of people who pursued biology education research (i.e., biology education research faculty). 

While the recruitment methods are mentioned, the sampling technique is not clear.

*We add our sampling methods to the text, as well as the Whitehead and White

---

## [Editor Report · Decision Letter 1]

4 Oct 2024

“It let me merge my love of teaching with research”: A qualitative investigation of the career pathways of biology education researchers

PONE-D-24-22006R1

Dear Dr. Driessen,

We’re pleased to inform you that your manuscript has been judged scientifically suitable for publication and will be formally accepted for publication once it meets all outstanding technical requirements.

Kind regards,

Yolanda Malele-Kolisa, BDS, MPH, MDent, PhD

Academic Editor

PLOS ONE
---

## [Editor Report · Acceptance letter]

8 Oct 2024

PONE-D-24-22006R1 

PLOS ONE

Dear Dr. Driessen, 

I'm pleased to inform you that your manuscript has been deemed suitable for publication in PLOS ONE. Congratulations! Your manuscript is now being handed over to our production team.

Kind regards, 

on behalf of

Prof Yolanda Malele-Kolisa 

Academic Editor

PLOS ONE